# *Lactiplantibacillus plantarum DSM20174* Attenuates the Progression of Non-Alcoholic Fatty Liver Disease by Modulating Gut Microbiota, Improving Metabolic Risk Factors, and Attenuating Adipose Inflammation

**DOI:** 10.3390/nu14245212

**Published:** 2022-12-07

**Authors:** José I. Riezu-Boj, Miguel Barajas, Tania Pérez-Sánchez, María J. Pajares, Miriam Araña, Fermín I. Milagro, Raquel Urtasun

**Affiliations:** 1Department of Nutrition, Food Sciences, and Physiology, University of Navarra, 31008 Pamplona, Spain; 2Center for Nutrition Research, University of Navarra, 31008 Pamplona, Spain; 3Navarra Institute for Health Research (IdiSNA), 31008 Pamplona, Spain; 4Biochemistry Area, Department of Health Science, Public University of Navarre, 31008 Pamplona, Spain; 5Centro de Investigación Biomédica en Red Fisiopatología de la Obesidad y Nutrición (CIBERobn), Instituto de Salud Carlos III, 28029 Madrid, Spain

**Keywords:** fatty liver progression, probiotics, intestinal microbiota, fat tissue inflammation

## Abstract

Non-alcoholic fatty liver disease (NAFLD) is the most common cause of chronic liver disease, reaching epidemic proportions worldwide. Targeting the gut–adipose tissue–liver axis by modulating the gut microbiota can be a promising therapeutic approach in NAFLD. *Lactiplantibacillus plantarum*, a potent lactic-acid-producing bacterium, has been shown to attenuate NAFLD. However, to our knowledge, the possible effect of the *Lactiplantibacillus plantarum* strain *DSM20174* (*L.p. DSM20174*) on the gut–adipose tissue axis, diminishing inflammatory mediators as fuel for NAFLD progression, is still unknown. Using a NAFLD mouse model fed a high-fat, high-fructose (HFHF) diet for 10 weeks, we show that *L.p DSM20174* supplementation of HFHF mice prevented weight gain, improved glucose and lipid homeostasis, and reduced white adipose inflammation and NAFLD progression. Furthermore, 16S rRNA gene sequencing of the faecal microbiota suggested that treatment of HFHF-fed mice with *L.p DSM20174* changed the diversity and altered specific bacterial taxa at the levels of family, genus, and species in the gut microbiota. In conclusion, the beneficial effects of *L.p DSM20174* in preventing fatty liver progression may be related to modulations in the composition and potential function of gut microbiota associated with lower metabolic risk factors and a reduced M1-like/M2-like ratio of macrophages and proinflammatory cytokine expression in white adipose tissue and liver.

## 1. Introduction

Non-alcoholic fatty liver disease (NAFLD) is reaching epidemic proportions worldwide, being the leading cause of chronic liver disease [1]. Excessive accumulation of triglycerides within hepatocytes characterises steatosis. However, the presence of additional histological abnormalities, such as lobular inflammation and hepatocyte injury, may determinate progression to a more severe stage termed as non-alcoholic steatohepatitis (NASH). NASH can lead to serious liver damage, including cirrhosis and hepatocarcinoma (HCC) [2]. Although individuals with steatosis have a benign clinical course, a wide range of affected patients (10–30%) develop NASH over time, with a poor long-term prognosis [3]. However, despite significant efforts in recent years to discover new drugs, no approved therapies are yet available to treat NASH. NAFLD is closely related to metabolic syndrome, insulin resistance, and type 2 diabetes mellitus [4]. Although overweight and obesity are the main drivers to develop metabolic syndrome and NAFLD, not all patients with obesity progress to metabolic fatty liver disease.

The underlying mechanism for the development and progression of NAFLD is complex and multifactorial [5,6]. Pathological connections between several organs, including an intricate triangle interplay between the gut, adipose tissue, and liver, seems to be involved [7,8]. In the liver, uncontrollable lipotoxicity facilitates reactive oxygen species (ROS) formation, which ultimately activates a cascade of apoptotic signals, thus promoting liver injury, inflammation, and fibrogenesis [9]. Moreover, the liver integrates gut- and adipose-tissue-derived signals, contributing to NASH progression [10]. Obesity and diet composition may affect gut microbial homeostasis and the intestinal integrity barrier, thus favouring gut–liver dysfunction. For example, disturbed intestinal permeability favours the influx of bacterial products and pathogen-associated molecular patterns (PAMPs) into the gut–liver axis, contributing to inflammation that further triggers NASH progression [11,12]. Gut-bacteria-derived lipopolysaccharide (LPS) and other PAMPs may also be key drivers of adipose tissue inflammation. Thus, metabolically unhealthy adipose tissue is characterised by an increase in proinflammatory macrophages also known as “M1” along with a decrease in anti-inflammatory macrophages or “M2” [13]. M1 macrophages secrete pro-inflammatory cytokines that ultimately inhibit insulin signalling in adipocytes, contributing to systemic insulin resistance. In contrast, M2 macrophages secrete anti-inflammatory cytokines that maintain insulin responsiveness [14,15]. A pivotal role for macrophages in the crosstalk between adipose tissue and NAFLD has been proposed. In this regard, several studies point to the fact that adipose tissue inflammation probably precedes liver inflammation. Supporting this idea, mouse models of NASH show increased expression of macrophage and inflammatory genes in adipose tissue earlier that in the liver [16,17]. In addition, intervention of NAFLD by removing the inflamed white adipose tissue (WAT) prevented NASH progression in animal models [18]. On the other hand, macrophages from obese visceral adipose tissue transplanted into mice amplified hepatic inflammation through increased hepatic macrophage infiltration [19]. In recent years, increasingly more attention has been paid to the role of gut microecology on diseases. In this regard, scientific evidence shows that beneficial bacteria or probiotics can modulate the gut microbiome. They are considered as a novel approach to prevent and treat different metabolic diseases such as NAFLD [20]. *Lactobacillus plantarum* (now known as *Lactiplantibacillus plantarum*, *L.p*.) is considered a potential probiotic and is present in various environments, such as soil and the human gut [21]. Different strains of *L.p*. have demonstrated beneficial effects on NAFLD through various mechanisms such as restoration of gut microbiota [22] and intestinal permeability [23], also reducing the novo fatty acid synthesis [21], oxidant stress [21,24], metabolic endotoxemia [25], and liver inflammation [23]. However, to our knowledge, the potential effect of the strain type *L.p. DSM20174* on NAFLD progression is still unknown. Therefore, the present study aimed to investigate the beneficial effect of *L.p DSM20174* on the modulation of gut microbial composition and its possible relationship with metabolic risk factors and adipose tissue inflammation in a mouse model of high-fat, high-fructose diet (HFHF)-induced NAFLD.

## 2. Materials and Methods

### 2.1. Animal and Experimental Design

The animal care and use protocol was approved by the Institutional Committee on Care and Use of Laboratory Animals (CEEA, University of Navarra, Protocol number: 017-20). All animal procedures were performed in accordance with the guidelines on animal welfare of the European Directive 2010/63/EEC and the RD 53/2013. All animal handling and methods complied with the ARRIVE guidelines. The 8-week-old male C57BL/6 mice were purchased from Envigo Laboratories (Horst, The Netherlands). After two weeks of acclimation, the animals were randomly divided and allocated into three groups (*n* = 6 each). In the normal diet group (ND group), animals received a chow diet (2014 Tekland Global 14% Protein Rodent Maintenance Diet, Envigo, Horst, The Netherlands) and water. In the high-fat, high-fructose group (HFHF group), animals were fed with a high-fat diet rich in lard (60% kcal from fat Teklad Diet TD.06414, Envigo) and fructose (Sigma-Aldrich, St. Louis, MO, USA) at 10% in the drinking water for 10 weeks. The HFHF group plus *Lactobacillum plantarum DSM20174* involved ingesting L.p 39 (IAM 12477), a type strain isolated from a pickled cabbage from the Spanish Collection of Type Cultures (CECT) (reference CECT 748T), for 10 weeks (HFHF + L.p group).

The nutritional content of the experimental diets is shown in Appendix A. In the HFHF + L.p group, the probiotic was administrated at a daily dose of 1 × 10^9^ CFU per animal, mixed in with the experimental diet. At the end of the experimental period (10 weeks), animals were sacrificed by cervical dislocation after overnight fasting for the collection of tissues and blood samples.

### 2.2. Body Weight and Serum Biochemical Analysis

During the experimental study, body weight was measured weekly. Serum was obtained from blood samples after centrifugation at a maximum speed of 3.000 rpm for 15 min at 4 °C in a Beckman coulter centrifuge. The supernatant (serum) was collected and diluted 1/10 with saline for the determination of cholesterol, HDL cholesterol (HDL), triglycerides (TG), and non-esterified fatty acids (NEFA) on a Roche/Hitachi Cobas analyser system.

### 2.3. Fasting Blood Glucose and Tolerance Test

Mice were fasted overnight, with free access to water prior to the test, and blood samples were obtained from the tip of the tail vein. Glucose concentrations were monitored using a pre-calibrated glucometer (Accu-chek Aviva, Roche, Basel, Switzeland), and glycemia was recorded once a week. Serum C-peptide was assayed by the sandwich ELISA method using an Ultra-Sensitive Mouse C-peptide ELISA kit (Abynkek Biopharma, Derio, Spain). The glucose tolerance test (GTT) was assessed at the end of the experimental animal protocol.

Mice were fasted overnight, and 2 g/kg body weight of glucose was injected intraperitoneally. Measurement of blood glucose levels was performed at 0, 20, 40, 60, 90, and 120 min after glucose injection by collection of tail blood samples. The area under the curve (AUC) of glucose values was assessed for each group from 0 to 120 min. The Homeostasis Model Assessment of Insulin Resistance (HOMA-IR) by C-peptide was calculated by the formula: HOMA-IR = serum C-peptide (nmol L^−1^) × serum glucose (nmol L^−1^)/22.5 [26]. The product of fasting triglycerides and glucose (TYG index) is a commonly used insulin resistance marker related to metabolic disorders [27]. The TYG index was calculated following the formula: TYG index = ln [Fasting triglyceride (mg/dL) × Fasting glucose (mg/dL)]/2.

### 2.4. Liver and Adipose Histology and Morphometry

Liver and epididymal adipose tissue obtained from necropsy were fixed in 10% formalin for 24 h. The tissues were dehydrated with xylene, were paraffin embedded, and were processed for haematoxylin and eosin (H&E) staining. For immunohistochemical (IHC) analysis, sections of white adipose tissue (WAT) and liver were stained with a primary anti-macrophage CD68 antibody (Cell signalling) and F4/80 antibody (Millipore), respectively, and counterstained with haematoxylin. The slides were digitised using a histology slide scanner Amperio CS (Leica Biosystems). The morphometric estimation of hepatic steatosis and the semi-quantitative IHC of CD68 and F4/80 was analysed with ImageJ Fiji software. To quantify adipocyte number and areas in the adipose tissue, digital images were analysed with the Adiposoft software (CIMA, University of Navarra).

### 2.5. Quantitative Real-Time PCR (RT-qPCR)

Total RNA was extracted from the livers and adipose tissue of mice using an RNAeasy Mini Kit (Qiagen, Germantown, MD, USA). Then, cDNA synthesis (RT) was performed with RT Premix (Invitrogen). Quantitative real-time PCRs (qPCR) were performed with the IQ SYBR Green Supermix (Bio-Rad, Hercules, CA, USA) in a CFX96 Real-Time System (Bio-Rad, Hercules, CA, USA). Gene expression was normalised relative to that of the housekeeping gene RPLPO (ribosomal protein lateral stalk subunit PO). The primer pairs used are listed in Table 1. Heat map and hierarchical clustering of gene expression was calculated using CFX manager software version 1.5 (Bio-rad, Hercules, CA, USA).

### 2.6. Faecal Microbiome Analysis

Faecal samples were collected at the end of the study before the dissection of the mice, using an aseptic technique and immediately frozen at −80 °C for metagenomic analyses. Faecal DNA was extracted from the stool samples obtained from the three mice groups (*n* = 6 mice per group) at the end of the experimental animal protocol and sequenced on the MiSeq platform (Illumina, San Diego, CA, USA) in CIMA Labs Diagnostics (Pamplona, Spain). Bacterial DNA isolation was carried out with a Promega-Maxwell^®^ RSC equipment using the Maxwell RSC Fecal Microbiome DNA Kit (Promega Corporation, Madison, WI, USA). For each DNA sample, the V3–V4 hypervariable regions of the 16S rRNA gene were amplified using specific primers (Illumina). The 16S rRNA sequences were filtered following quality criteria of the processing pipeline LotuS (release 1.58) [28]. The pipeline includes UPARSE de novo sequence clustering for the identification of OTUs (operational taxonomic units) and their abundance matrix generation by similarities in DNA sequence (Appendix A). Finally, taxonomy was assigned using the Greengenes database. The OTUs not identified with this database were reanalysed by BLAST and the Ribosomal Database Project (RDP).

### 2.7. Statistical Analysis

Except for microbiota, statistical analyses were performed with GraphPad Prism 8.0 software. Data were expressed as the mean ± standard deviation (SD). The statistical analyses of body weight, serum glucose levels over time, and glucose tolerance test were performed by two-way repeated measures ANOVA followed by Bonferroni post hoc test. Comparisons among groups were tested by one-way analysis of variance (ANOVA) followed by Bonferroni post hoc test. Student’s t or Mann–Whitney U test was used for the two group comparisons once normality was calculated with a Shapiro–Wilk test. *p* < 0.05 was considered statistically significant. Microbiota data (as families, genera, and species) were analysed using the MicrobiomeAnalyst platform [29]. Alpha diversity of gut microbiota (observed genera and species) were analysed using a nonparametric test. Analysis of the β-diversity represented by principal coordinate analysis (PCoA) was calculated using the Bray–Curtis index and PERMANOVA test. Statistical differences in microbiota abundances between groups were tested by EdgeR (Empirical Analysis for Digital Gene Expression in R) previously normalised using trimmed mean of values normalisation (TMM). The association between microbiota and underlying factors contributing to NAFLD was evaluated by Spearman correlation using Stata 16 separately in metabolic parameters and inflammatory gene expression corrected by FDR (considered significant when FDR < 0.05).

## 3. Results

### 3.1. Effect of Lactiplantibacillus plantarum DSM 20174 on Body Weight and Serum Lipids

To investigate the effects of *Lactiplantibacillus plantarum DSM 20174* (*L.p DSM20174*) on NAFLD, we established a murine model of NAFLD by feeding mice with an HFHF diet for ten weeks. HFHF feeding increased body weight significantly compared with the ND group (Figure 1a,b). Notably, *L.p DSM20174* supplementation significantly prevented body weight gain in the HFHF-fed mice at the end of the experiment (Figure 1a,b). Serum levels of total cholesterol, HDL, TG, and NEFAs were significantly increased in the HFHF group (Figure 1c). After ten weeks, *L.p DSM20174* supplementation in HFHF-fed mice significantly reduced serum TG, total cholesterol, and NEFAs compared with the HFHF group (Figure 1c). There were no significant differences in the serum levels of HDL among the HFHF groups (Figure 1c).

### 3.2. Effect of Lactiplantibacillus plantarum DSM 20174 on Glucose Metabolism

After ten weeks of HFHF feeding, fasting glucose, c-peptide levels, and HOMA-IR in the HFHF group were found to be significantly higher than in the ND group (Figure 2a,b). In contrast, glucose, serum c-peptide levels, and HOMA-IR remained lower in the HFHF group supplemented with *L.p DSM20174* (Figure 2a,b). As expected, mice fed the HFHF diet showed significantly impaired glucose tolerance and insulin sensitivity compared to those in the ND group, as measured by the glucose tolerance test (GTT) (Figure 2c). In the GTT, the HFHF group had significantly higher blood glucose levels at 0, 40, 90, and 120 min compared to the ND group. Moreover, *L.p DSM20174* administration significantly prevented the impaired glucose tolerance caused by the HFHF diet (Figure 2c). Similarly, the AUC derived from the GTT was significantly larger for the HFHF group compared with the rest of groups (Figure 2c). HFHF consumption increased the TYG index, with *L.p DSM20174* significantly decreasing this index (Figure 2d).

### 3.3. Effect of Lactiplantibacillus plantarum DSM 20174 on Adipose Tissue Inflammation

Histological examination of WAT showed that HFHF diet increased the size of adipocytes at the end of the experiment (Figure 3a). This trend was further confirmed by quantification of adipocyte area using automated software, which showed that the average area was 4846 ± 650 µm^2^ for mice on an HFHF diet, while that for those on ND was only 1159 ± 475 µm^2^. Supplementation of HFHF with *L.p DSM20174* had no significant effect on mean adipocyte size or surface area (4529 ± 757 µm^2^) (Figure 3b). Expansion of WAT and hypertrophy of adipocytes are usually associated with adipose inflammation. WAT inflammation was defined by the presence of dead adipocytes surrounded by macrophages forming crown-like structures (CLS). After ten weeks of HFHF diet exposure, the CLS were present in the WAT samples. Interestingly, supplementation with *L.p DSM20174* decreased the number of CLS (Figure 3a, arrowheads). Moreover, aggregates of CD68-positive macrophages were significantly higher in CLS surrounding individual adipocytes of the HFHF group as a hallmark of localised chronic inflammation. On the contrary, the WAT of the HFHF mice supplemented with *L.p DSM20174* showed significantly fewer CD68-positive macrophage rings around the adipocytes (Figure 3c). To further confirm the presence of adipose tissue inflammation, we studied the expression of various macrophage and inflammatory genes. Adipose tissue macrophages (ATMs) were classified as M1 or “classically activated” and M2 or “alternatively activated” macrophages. HFHF diet alters the balance of ATMs towards the pro-inflammatory M1 phenotype. The expression of pan-macrophage marker genes (F4/80 and CD68) and M1-associated genes (MCP1, CD44 and CD11c) increased remarkably in the HFHF group. However, L.p *DSM20174* supplementation significantly attenuated F4/80, MCP1, and CD11c mRNA and decreased CD68 and CD44 mRNA expression (Figure 3d). Moreover, *L.p DSM20174* treatment significantly upregulated the expression of M2-associated genes, including CHI3L3 and RETNLA. *L.p DSM20174* supplementation also decreased the gene expression of Th1 pro-inflammatory cytokines (IL1B and IL6) and increased Th2 anti-inflammatory factors (IL4 and IL5) in the WAT of the HFHF-fed mice. Previous works demonstrated that pro-inflammatory cytokine secretion compromised adipogenesis by blocking the PPARγ transcriptional network in adipose tissue [30,31]. Our results are consistent with these findings and showed less PPARγ and SREBP gene expression in the adipose tissue of the HFHF-fed mice compared with those in the ND group. However, *L.p DSM20174* supplementation ameliorated the downregulation of those genes in HFHF-fed mice (Figure 3d).

### 3.4. Effect of Lactiplantibacillus plantarum DSM 20174 on Hepatic Lipid Acumulation

Scientific evidence has shown that low-grade inflammation causes adipose tissue dysfunction. These alterations have been associated with local and systemic insulin resistance and lipid deposition in non-adipose organs such as the liver [28,29]. Our data indicate that HFHF diet consumption for ten weeks leads to development of NAFLD in mice. Histological analysis of the liver sections showed no signs of steatosis in the ND group. The liver of HFHF-fed mice accumulated extensive microvesicular steatosis and showed a slight increase in portal and lobular inflammation (Figure 4a). However, *L.p DSM20174* supplementation significantly prevented HFHF-induced steatosis and lobular inflammation (Figure 4a,b). We also performed immunostaining for F4/80 as a monocyte-macrophage marker of hepatic infiltration and distribution in the liver. In the ND and HFHF group supplemented with *L.p DSM20174*, macrophages showed a scattered distribution in the liver. On the other hand, in the HFHF group, some macrophages aggregated to surround hepatocytes with large lipid droplets, a phenomenon previously termed “hepatic CLS” [30]. The number of F4/80-positive hepatic CLS was significantly increased in the HFHF group (Figure 4b). To confirm lipid accumulation and hepatic inflammatory status, we analysed the expression of several genes involved in lipid metabolism and inflammation in the liver. Our results showed a significant increase in the mRNA expression of adipogenic (PPARγ, CD36), lipogenic (FASN), and fatty acid β-oxidation (CPT1) genes in the HFHF group compared to the ND and HFHF + L.p groups. The relative gene expression of IL6, TLR4, and MCP1 was significantly lower in the *L.p DSM20174* treatment group compared to the HFHF group. No significant differences were found in F4/80 gene expression (Figure 4c). Hierarchical cluster analysis of these genes grouped HFHF separately from the rest of the groups (Figure 4d).

### 3.5. Effect of Lactiplantibacillus plantarum DSM 20174 on the Diversity of the Intestinal Microbiota

Recent pieces of evidence have demonstrated that gut microbiota might be a potential target for treating obesity and related metabolic diseases, such as NAFLD [32]. Nonetheless, there is growing evidence that some dietary substances, especially probiotics and prebiotics, can modulate gut microbiota.

This study examined the effect of *L.p DSM20174* on the composition and diversity of gut microbiota of HFHF-fed mice. HFHF groups showed a significant increase in Firmicutes/Bacteroidetes ratio (ND = 0.47 ± 0.24; HFHF = 1.52 ± 1.06 and HFHF + L.p = 1.44 ± 1.09). The α-diversity of gut microbiota (measured as the number of unique taxa observed in each sample) was significantly higher in the HFHF group supplemented with *L.p DSM20174* compared to ND and HFHF groups in terms of genera (Figure 5(aA)) and species (Figure 5(aB)). Analysis of the β-diversity represented by principal coordinate analysis (PCoA) based on Bray–Curtis dissimilarity of the genus profiles showed significant compositional differences among the three groups (Figure 5b). The results were statistically validated by the pairwise PERMANOVA *p*-value of <0.001.

### 3.6. Effect of Lactiplantibacillus plantarum DSM 20174 on Gut Microflora

Using the EdgeR statistical method, we analysed the differential abundance in gut microbiota composition among the three groups. We have only on those taxa (families, genera, and species) that present significant differences between HFHF and HFHF + L.p (Figure 6). At the family level, in the HFHF group, the relative abundance of Christenseneliaceae (belonging to the phylum Firmicutes) was significantly higher than in the ND group.

Interestingly, *L.p DSM20174* supplementation in the HFHF diet reduced the relative abundance of this family (Figure 6a). At the genus level, in the HFHF group, the relative abundance of *Christensenella* and *Phocaicola* was higher than in the ND group. In contrast, the relative abundance of *Acetatifactor*, *Duncaniella*, *Monoglobus* and *Lawsonibacter* was lower in the HFHF group compared to the ND group. Interestingly, *L.p DSM20174* supplementation partially prevented the HFHF-diet-induced changes in the relative abundance of these six genera (Figure 6b). At the species level, *L.p DSM20174* supplementation significantly reduced *Ruminococcus torques* levels compared to the HFHF and ND groups (Figure 6c). Finally, *Lactiplantibacillus plantarum* spp. was detected only in faecal samples of mice supplemented with *L.p DSM20174*.

### 3.7. Association of Some Microbial Taxa with Urderlying Factors Contributing to NAFLD

Spearman’s correlation coefficient was used to analyse the relations between significant intestinal bacteria changes in HFHF-fed mice and metabolic risk factors (Figure 7a) or inflammatory genes (Figure 7b) associated with NAFLD.

First, the association between metabolic risk factors associated with NAFLD and gut microbiota changes was studied. As shown in Figure 7a, body weight gain, triglyceride levels, and the TYG index showed a positive and strong association with the Christensenellaceae family, *Christensenella* genus, and *Ruminococcus torques*. In contrast, *Monoglobus* levels were negatively associated with the TYG index and blood glucose, *Lactiplantibacillus plantarum* abundance was negatively associated with the TYG index and triglyceride levels, and *Acetatifactor* and *Lawsonibacter* abundances were negatively correlated with body weight gain.

Second, it is well known that NAFLD progression is associated with impaired function of genes involved in WAT and hepatic inflammation. To comprehensively analyse the relations between inflammatory gene expression and gut microbiota, another correlation matrix was generated. The expression of pan- (F4/80) and M1 (MCP1, CD11c) macrophage markers and proinflammatory (IL1B, IL6) genes in WAT and liver were positively associated with Christensenellaceae, *Christensenella*, *Phocaicola*, and *Ruminococcus* torque abundance. In contrast, the same microbial taxa correlated negatively with the expression of a M2-macrophage-related gene (RENTLA) and with the IL5 gene in WAT. On the other hand, *Acetatifactor* and *Lawsonibacter* were negatively associated with IL1B gene expression in WAT but positively correlated with RENTLA expression. *Lactiplantibacillus plantarum DSM20174* correlated negatively with the expression of pan- (F4/80) and M1 (MCP1, CD11c) macrophage markers and pro-inflammatory (IL1B, IL6) genes in WAT and liver but positively correlated with the expression of IL5 in WAT (Figure 7b).

## 4. Discussion

*Lactiplantibacillus plantarum* is a lactic-acid-producing bacterium found in many fermented foods. Different strains of *Lactiplantibacillus plantarum* reduce metabolic abnormalities, including obesity-induced hepatic steatosis [21,22,23,24,25]. However, the role of *Lactiplantibacillus plantarum DSM 20174* (*L.p DSM20174*) in preventing NAFLD progression has not yet been demonstrated. Our results showed that *L.p DSM20174* supplementation could improve diet-induced obesity, ameliorate glucose and lipid metabolic disturbances, and prevent hepatic steatosis in a diet-induced NAFLD mouse model. The beneficial effects of *L.p DSM20174* can be explained by different mechanisms, including the modulation of gut microbiota composition associated with lower metabolic risk factors and a reduced M1/M2 macrophage ratio in WAT. Both mechanisms could play essential roles in preventing fatty liver progression in a diet-induced obese model of NAFLD in mice.


**Association of the gut microbiota with metabolic risk factors in NAFLD**


Regarding the first mechanism, diet is considered the primary driver of gut microbiome community structure. Beta-diversity showed that HFHF feeding significantly impacted gut microbial composition and significantly altered the relative abundance of bacteria consistent with previous reports [33]. *L.p DSM20174* supplementation significantly restored the relative abundance values of specific bacterial taxa to near normal levels.

Moreover, it increased the alpha diversity of the gut microbiota (associated with a healthy state of the individual). Our results, in agreement with previous studies in mice, showed that Christensenellaceae and *Christensenella* increased their relative abundance in the gut in response to HFD feeding [34,35,36]. The correlation matrix analyses also showed that Christensenellaceae and *Christensenella* positively correlated with body weight gain, triglyceride levels, and the TYG index, a marker of insulin resistance with high specificity in identifying metabolic syndrome [27]. *L.p DSM20174* supplementation decreased the relative abundance of Christensenellaceae and *Christensenella* and reduced the body weight gain, serum lipid levels, and the TYG index, decreasing the likelihood of suffering from insulin resistance and metabolic syndrome in an HFFD-fed mouse model. In addition, *L.p DSM20174* significantly reduced the relative abundance of *Ruminococcus torques*, positively associated with triglyceride levels and TYG index. Consistent with this, previous researchers showed that a HFHF diet resulted in dysbiosis with increased *Bacteroides* spp. and *Ruminococcus torques* [37]. Furthermore, a recent study demonstrated that *Ruminococcus torques* is one of the most predictive bacterial species for obesity [38]. Our research also showed that the relative abundance of *Acetatifactor* and *Lawsinobacter* correlated negatively with increased body weight. *L.p DSM20174* administration specifically increased the relative abundance of *Acetatifactor* to levels even higher than usual. In the literature, the *Acetatifactor* genus has been described to activate the bile acid membrane receptor TGR4 and, in this way, stimulate GLP-1 secretion, which improves obesity [39] and, consequently, liver function and tolerance to insulin and glucose [40]. The present work also describes that the abundance of *Monoglobus* and *Lactiplantibacillus plantarum* spp. increases after *L.p DSM20174* supplementation in HFHF mice. Interestingly, *Monoglobus* and *Lactiplantibacillus plantarum* spp. correlated negatively with an altered lipid profile, glucose levels, and TYG index. *Monoglogus pectinilyticus* spp., which has a highly specialised glycobiome for pectin degradation, is present in humans with a high intake of dietary fibre and pectin [41] and is more common in the gut microbiome of mice fed a very low-calorie diet [42]. Several studies have demonstrated the beneficial effect of oral treatments with different strains of *Lactiplantibacillus plantarum* in lowering blood triglyceride levels [43,44]. However, to our knowledge, this is the first study to show a negative correlation between the relative abundance of *Lactiplantibacillus plantarum* spp. in the gut and an altered lipid profile.

**Association of the gut microbiota with adipose tissue inflammation**.

Regarding the second mechanism, our results demonstrate that *L.p DSM20174* plays an essential role in modulating the inflammatory response in the white adipose tissue by shifting adipose M1/M2 polarisation toward the M2 phenotype. The M2 activation inflammatory program in adipose tissue may prevent disease progression locally and in the liver. In the literature, the importance of ATMs in NAFLD was further corroborated in humans, as both the adipose tissue expression of pro-inflammatory genes and the number of ATMs were associated with the progression of NAFLD [45]. The influence of *L.p DSM20174* on the gut microbiota may be involved in modulating the inflammatory response in adipose tissue. In this sense, increased abundance of *Acetatifactor*, *Monoglobus*, *Lawsonibacter*, and *Lactiplantibacillus plantarum* spp., together with decreased abundance of Christensenellaceae, *Christensenella*, *Phocaeicola*, and *Ruminococcus torques*, positively correlated with a reduced M1/M2 ratio of macrophages, increased the expression of the *Il5* gene in adipose tissue, and decreased expression of pro-inflammatory markers in the liver. The role of *L.p* in triggering macrophages to the M2 phenotype has been previously described in the literature. L.p-derived extracellular vesicles induced macrophage polarisation towards the M2 state in human monocytic cells [46]. Furthermore, *Lactiplantibacillus plantarum CLP-0611* polarises M1- to M2-like macrophages, thereby ameliorating colitis in mouse models [47]. However, further studies are necessary to elucidate the underlying mechanism of L.p in switching M1/M2 phenotypes in adipose tissue.

In summary, our results demonstrate that *L.p DSM20174* supplementation could prevent HFHF-induced NAFLD in mice. The potential mechanism involved in this phenomenon could be due to the observed changes in the abundance of functionally relevant gut bacteria that could be directly related with decreased metabolic risk factors and the M1/M2 inflammatory state in adipose tissue.

## Figures and Tables

**Figure 1 nutrients-14-05212-f001:**
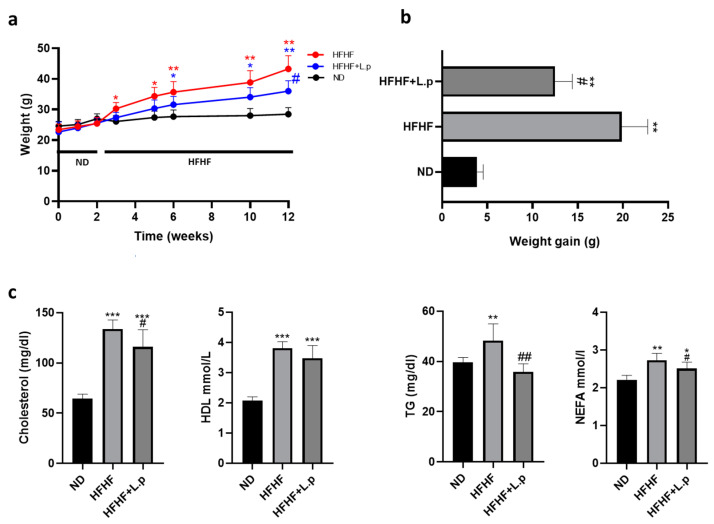
Body weight, weight gain, and serum lipid profile in HFHF mice. (**a**) Changes of body weight during the animal experimental study. (**b**) Differences of body weight gain between groups. (**c**) Levels of total cholesterol, HDL, triglycerides (TG), and non-esterified fatty acids (NEFA). Data are expressed as mean ± SEM. * *p*  <  0.05, ** *p*  <  0.01, *** *p*  <  0.001 vs. ND group: # *p*  <  0.05, ## *p*  <  0.01 vs. HFHF diet.

**Figure 2 nutrients-14-05212-f002:**
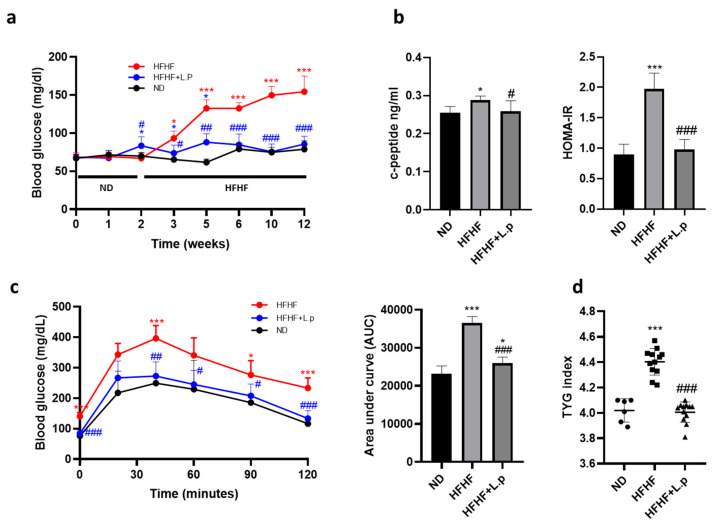
The effect of *L.p DSM20174* supplementation on glucose homeostasis and TYG index. (**a**) Fasting blood glucose measured at several time points during the experimental procedure. (**b**) Serum c-peptide and HOMA-IR index. (**c**) Blood glucose levels in response to glucose tolerance test (GTT). Effect on area under de curve (AUC). (**d**) Triglyceride glucose (TYG) index. Data are expressed as mean ± SEM. * *p* < 0.05, *** *p* < 0.001 vs. ND; # *p* < 0.05, ## *p* < 0.01, ### *p* < 0.001 vs. HFHF.

**Figure 3 nutrients-14-05212-f003:**
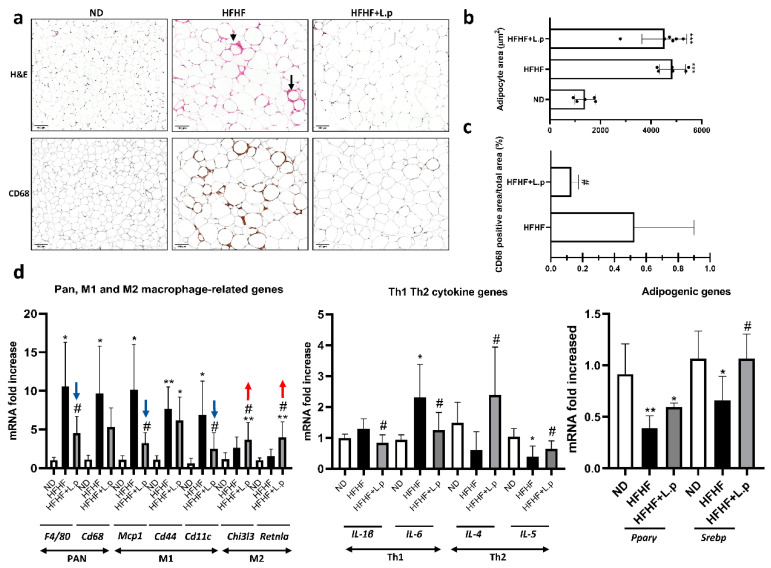
*L.p DSM20174* supplementation reduced white adipose tissue inflammation in HFHF-fed mice. (**a**) Images of haematoxylin and eosin (H&E)-stained sections of white adipose tissue (WAT). Arrowheads indicate crown-like structures (CLS). Representative immunostaining for CD-68. (**b**) Mean adipocyte area. (**c**) CD-68 positive area (%). (**d**) Relative gene expression of the Pan, M1, and M2 macrophage markers; Th1 and Th2 cytokines; and adipogenic genes. All genes were normalised to expression of RPLPO. Blue arrows indicate decreased M1 macrophage marker expression, whereas red arrows show increased M2 macrophage marker expression. Data are expressed as mean ± SEM. * *p* < 0.05, ** *p* < 0.01, *** *p* < 0.001 vs. ND; # *p* < 0.05 vs. HFHF.

**Figure 4 nutrients-14-05212-f004:**
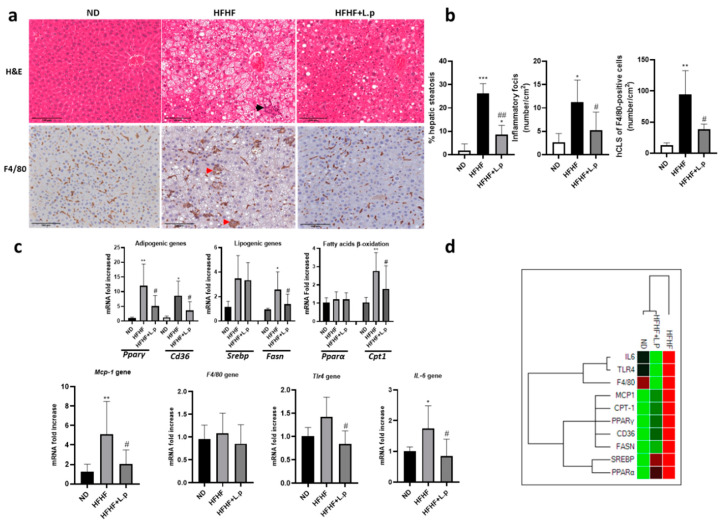
Effect of *L.p DSM20174* on liver lipid content and inflammation. (**a**) Images of haematoxylin and eosin (H&E)-stained sections of liver. Black arrowheads indicate lobular inflammation. Representative immunostaining for F4/80. Red arrowheads indicate hepatic crown-like structures (h-CLS). (**b**) Percentage of hepatic steatosis. The number of inflammatory foci per cm^2^. The number of hCLS F4/80 positive cells per cm^2^. (**c**) Fold changes in expression of genes related to adipogenic, lipogenic, fatty acid β-oxidation functions, and liver inflammation. All genes were normalised to expression of RPLPO. (**d**) Hierarchical heatmap clustering of lipid metabolism and inflammatory gene expression in liver mice. Red and green shading represent higher and lower relative expression levels, respectively. Data are expressed as mean ± SEM. * *p* < 0.05, ** *p* < 0.01, *** *p* < 0.001 vs. ND; # *p* < 0.05, ## *p* < 0.01 vs. HFHF.

**Figure 5 nutrients-14-05212-f005:**
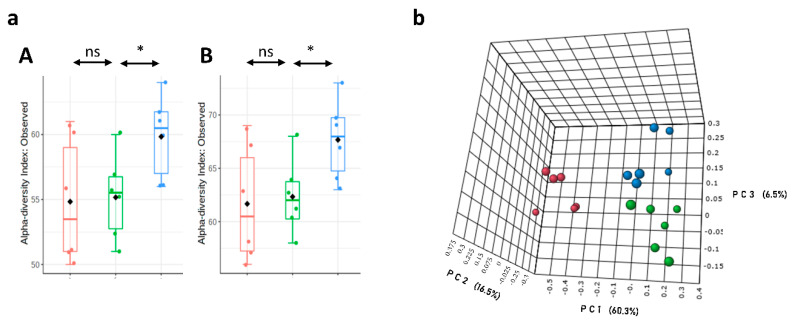
*L.p DSM20174* modified the diversity of gut microbiota. (**a**) Alpha diversity, measured by observed genera (**A**) and species (**B**) in faecal samples of ND, HFHF, and HFHF + L.p mice. Mean values and interquartile ranges are indicated in the plots. (**b**) Beta-diversity (mean of different genera within mice). * Significant difference between HFHF and HFHF + L.p (*p* < 0.05). Red boxes and circles represent the ND group, green boxes and circles represent the HFHF group, and blue boxes and circles represent the HFHF + L.p group (*n* = 6 mice per each group).

**Figure 6 nutrients-14-05212-f006:**
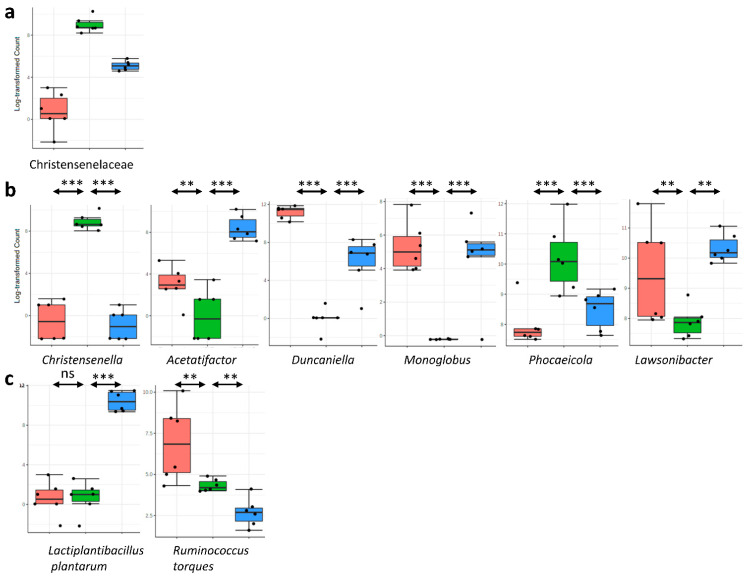
*L.p DSM20174* supplementation altered the composition of gut microbiota in HFHF-fed mice. The relative abundance at (**a**) family, (**b**) genus, and (**c**) species levels in the faecal samples from the three groups. Mean values and interquartile ranges are indicated in the plots. ** *p* < 0.01, *** *p* < 0.001. Red boxes represent the ND group, green boxes represent the HFHF group, and blue boxes represent the HFHF + L.p group.

**Figure 7 nutrients-14-05212-f007:**
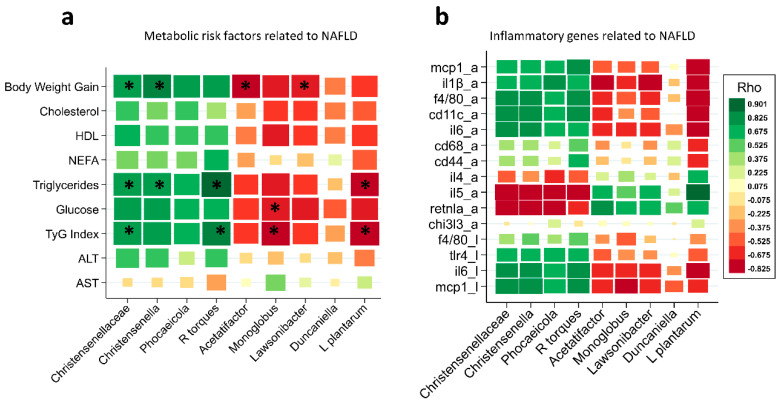
Correlation of gut microbiota changes with underlying factors related to NAFLD: (**a**) metabolic risk factors related to NAFLD; (**b**) inflammatory genes related to NAFLD; “a” in adipose tissue: “l” in liver. Spearman correlation coefficients (Rho) and the corresponding *p*-values were calculated on the basis of comparisons of the relative abundance from species, genera, and families, as well as the underlying factors related to NAFLD. The green colour indicates a positive correlation coefficient, and the red colour represents a negative coefficient (* FDR *p* < 0.05).

**Table 1 nutrients-14-05212-t001:** Primer sequences for qPCR.

Gene	Forward Primer 5′-3′	Reverse Primer 5′-3′	Product Size (bp)	Annealing Temp (°)
RPLPO	AACATCTCCCCCTTCTCCTT	GAAGGCCTTGACCTTTTCAG	270	62
PPARG	GCTGTTATGGGTGAAACTCTG	GAATAATAAGGTGGAGATGCAGG	356	60
CD36	CACAGCTGCCTTCTGAAATGTGTGG	TTTCTACGTGGCCCGGTTCTAATTC	171	60
PPARα	ACTGGTAGTCTGCAAAACCAAA	AGAGCCCCATCTGTCCTCTC	153	60
SREBP	CACTTCATCAAGGCAGACTC	CGGTAGCGCTTCTCAATGGC	284	60
FAS	AGCCATGGAGGAGGTGGTGAT	GTGTGCCTGCTTGGGGTGGAC	223	60
IL1B	TCGCTCAGGGTCACAAGAAA	CATCAGAGGCAAGGAGGAAAAC	73	60
IL6	ACAAGTCGGAGGCTTAATTACACAT	TTGCCATTGCACAACTCTTTTC	72	60
CPT1	TCTAGGCAATGCCGTTCAC	GAGCACATGGGCACCATAC	99	60
F4/80	CCTGGACGAATCCTGTGAAG	GGTGGGACCACAGAGAGTTG	64	60
CD68	GTGTCTGATCTTGCTAGGACC	TGTGCTTTCTGTGGCTGTAG	118	60
TLR4	GCCTTTCAGGGAATTAAGCTCC	AGATCAACCGATGGACGTGTAA	115	60
MCP1	TGATCCCAATGAGTAGGCTGGAG	ATGTCTGGACCCATTCCTTCTTG	132	62
CD44	CTGGATCAGGCATTGATGATG	GCCATCCTGGTGGTTGTCTG	157	60
CD11c	GATGGCTCGGGTAGCATCAG	TGAGGACCTTGGTGGCATCT	295	60
RETNLA	CTGGGATGACTGCTACTGGG	CAGTGGTCCAGTCAACGAGTA	108	60
CHI3L3	CCAGCAGAAGCTCTCCAGAAG	TCAGCTGGTAGGAAGATCCCA	161	60
IL4	ACGAAGAACACCACAGAG	TGATGTGGACTTGGACTC	195	60
IL5	TGTTGACAAGCAATGAGACGATGA	AATAGCATTTCCACAGTACCCCCA	136	60

## Data Availability

Public databases used in this study included Greengenes (http://greengenes.lbl.gov), the Ribosomal Database Project (RDP; http://rdp.cme.msu.edu), and the NCBI (http://www.ncbi.nlm.nih.gov/taxonomy) accessed on 12 September 2022. The datasets generated or analysed during the current study are available on request from the corresponding author.

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
