# Peer review of "Lactiplantibacillus plantarum DSM20174 Attenuates the Progression of Non-Alcoholic Fatty Liver Disease by Modulating Gut Microbiota, Improving Metabolic Risk Factors, and Attenuating Adipose Inflammation"

_nutrients, 2022, doi:10.3390/nu14245212_

Round 1

Reviewer 1 Report

In this paper, the authors studied the effects of Lactiplantibacillus plantarum DSM20174 on the progression of non-alcoholic fatty liver disease. The authors systematically study this topic from the perspectives of changes in gut microbes and steatosis in adipose tissue. The trials were well designed, the results were reliable, the discussions were well conducted, and the conclusions were pertinent. I think it's a good study, but there are still some issues worth paying attention to.

First of all, the author should fully describe the characteristics of this strain of Lactiplantibacillus plantarum and how it differs from others in previous studies.

The most direct impact of probiotics added to feed is the structure of the intestinal flora and intestinal health. Changes in the gut may further affect other tissues by metabolites or secretory function of the intestine. Therefore, if the authors in this study can analyze the immune and antioxidant properties of the intestine, it is very beneficial to explain the hypothetical second mechanism.

Table 1. product size and annealing temperature should be given.

Metabolic dysfunction-associated fatty liver disease (MAFLD), previously known as non-alcoholic fatty liver disease (NAFLD), has been proposed as a better definition with “positive criteria” to identify the liver disease associated with known metabolic dysfunction, based on evidence of hepatic steatosis, in addition to one of the following three criteria, namely overweight/obesity, presence of type 2 diabetes mellitus, or evidence of metabolic dysregulation. (https://doi.org/10.3390/ijms22031458)

Author Response

Response to Reviewer 1:
In this paper, the authors studied the effects of Lactiplantibacillus plantarum DSM20174 on the progression of non-alcoholic fatty liver disease. The authors systematically study this topic from the perspectives of changes in gut microbes and steatosis in adipose tissue. The trials were well designed, the results were reliable, the discussions were well conducted, and the conclusions were pertinent. I think it's a good study, but there are still some issues worth paying attention to.
We thank the reviewer for his/her valuable comments. We agree with all of them. We have addressed these comments in the revised manuscript, highlighted in red, and have provided a detailed answer to every raised question below. 

Comment #1. First of all, the author should fully describe the characteristics of this strain of Lactiplantibacillus plantarum and how it differs from others in previous studies.
Answer: We thank the reviewer for the comment, and in order to address this question, the sentence “Lactiplantibacillus plantarum DSM20174, L.p 39 [IAM 12477] a type strain isolated from a pickled cabbage from the Spanish Collection of Type Cultures (CECT) (Reference CECT 748T)”(https://www.uv.es/uvweb/coleccion-espanola-cultivos-tipo/es/catalogo-cepas/medios-cultivo/buscador-cepas-1285892802374.html) has been added to the text (lines 106-108 in the revised version). 
This specific strain of Lactiplantibacillus plantarum has been used in previous studies carried out by our laboratory that have shown its ability to control blood glucose in a mouse model fed with a hypercaloric diet and supplemented with apple cider vinegar (data not published). Although different research groups have used other strains with similar activity to the strain studied by our laboratory, none of them have demonstrated their protection against progression of non-alcoholic fatty liver disease though the modulation of gut microbiota associated with lower metabolic risk factors and a reduced M1-like /M2-like ratio of macrophages and proinflammatory cytokine expression in white adipose tissue and liver. However, we do not know if the rest of the published lactiplantibacillus strains develop this function through the mechanism described in our work.

Comment #2. The most direct impact of probiotics added to feed is the structure of the intestinal flora and intestinal health. Changes in the gut may further affect other tissues by metabolites or secretory function of the intestine. Therefore, if the authors in this study can analyze the immune and antioxidant properties of the intestine, it is very beneficial to explain the hypothetical second mechanism.
Answer: Thank you for pointing this out. Although we agree that this is an important consideration, it cannot be analyzed in this manuscript due to the lack of intestinal tissue samples and serum from the present animal experiment.
However, we appreciate the reviewer´s comment, as this is one of the major issues in the field of microbiota and intestinal health. In order to fully address this question, it is necessary to have systems that integrate “massive genome sequencing” of the microbiota present in mouse/human feces (preferably systems that allow samples to be obtained throughout different portions of the digestive tract), “metabolomics” for the analysis of metabolites derived from the microorganisms most frequently present in the microbiota, as well as “proteomics analysis” in blood samples that allow us to identify changes derived from the presence of a differentially expressed microbiota. The integration of the information obtained from these three systems would allow us to elucidate the role played by the microbiota and, by extension, the application of treatments based on probiotics such as our lactiplantibacillus sp., not only in the pathophysiology of different diseases, but also in the response of patients to pharmacological treatments. This is the step that we are carrying out and that we hope to be able to publish in the near future with the aim of better understanding the role played by the microbiota as another organ within our body.
We are also considering using a version of these Lactiplantibacillus-based probiotics in the form of postbiotics, so that we can determine if the presence of live bacteria is necessary to exert the observed effect on protection against NAFLD induced by a hypercaloric diet. In this sense, it has recently been shown that exosomes derived from lactobacilli could be responsible for the beneficial activity attributed to them with respect to their anti-inflammatory activity (doi: 10.1016/j.bbrc.2017.05.152). A representative example is the contribution of extracellular membrane vesicles in immune-dependent properties of Lactobacillus rhamnosus GG and L. reuteri DSM 17938 (doi: 10.1038/s41598-019-53576-6).
Comment #3. Table 1. product size and annealing temperature should be given.
Thanks for your recommendation. We now explicitly specify the product size (base pairs) and annealing temperature in Table 1 of the Material and Methods section on line 157 in the revised manuscript.
Comment #4. Metabolic dysfunction-associated fatty liver disease (MAFLD), previously known as non-alcoholic fatty liver disease (NAFLD), has been proposed as a better definition with “positive criteria” to identify the liver disease associated with known metabolic dysfunction, based on evidence of hepatic steatosis, in addition to one of the following three criteria, namely overweight/obesity, presence of type 2 diabetes mellitus, or evidence of metabolic dysregulation. (https://doi.org/10.3390/ijms22031458)
Answer: We appreciate the comment and the opportunity it offers us to introduce this novel concept of “Metabolic associated fatty liver disease (MAFLD)” proposed in 2020, since no publication that includes the use of lactiplantibacillus (or probiotics in general) has used this term to date. However, we do not know if this term is usable in the animal model used in the present manuscript since MAFLD definition is more practical for identifying patients with fatty liver disease with high risk of disease progression (DOI: 10.1111/liv.14548) and there is still some debate and some controversy regarding when to refer to NAFLD or MAFLD (DOI: 10.1111/liv.14739). On the other hand, there are authors who are already committed to the use of this new concept but to date always referring to patients (DOI: 10.1111/hepr.13706), not to experimental animal models. For this reason, we believe that it is still premature and daring to change the term NAFLD to MAFLD in our experimental model

Reviewer 2 Report

Please, find my comments and suggestion in the attached file.

Author Response

Response to Reviewer 2:

Manuscript entitled: Lactiplantibacillus plantarum DSM20174 attenuates the progression of non-alcoholic fatty liver disease by modulating gut microbiota, improving metabolic risk factors, and attenuating adipose inflammation. The paper is about induced obesity by feeding mice with high-fat-high-fructose diet during 10 weeks and counteracting negative effects related to glucose and lipid metabolic risks by supplementing Lactiplantibacillus plantarum DSM20174. Authors studied, glucose metabolism, lipid profiles, liver histology, gene expression in liver and adipose tissue and changes in gut microbiota composition. This study brings new knowledge about probiotic with potential to counteract diet-induced metabolic disturbances.

We thank the reviewer for his/her valuable comments. We agree with all of them. We have addressed these comments in the revised manuscript, highlighted in red, and have provided a detailed answer to every raised question below.  

ABSTRACT

Comment 1: Abstract: L.p, should be L.p. DSM 20174.

Answer: We agree with this comment. Therefore, we have changed “L.p” to “L.p. DSM 20174” (lines 25, 28, 31 and, 33 in the abstract revised version), but also, we have corrected it throughout the rest of the revised manuscript.

Comment 2: Key words: No repetition of the words from the title is suggested.

Answer: Authors agree. In the revised manuscript, we have replaced them with new key words (line 37).

INTRODUCTION

Comment 3: Line 56: What is ROS? Give full name before the abbreviation.

Answer: As suggested by the reviewer, we have included the expanded full name of reactive oxygen species (ROS) (line 56 in the revised version).

Comment 4: Line 88: What you mean by standard strain?.

Answer: We understand the reviewer´s question and have corrected the standard strain for type strain (line 88 in the revised version).

MATERIAL AND METHODS

Comments (A):

Comment: Chow diet: name and manufacture of the diet

Answer: Done. Normal Diet group (ND group), animals receiving chow diet (2014 Tekland Global 14% Protein Rodent Maintenance Diet, Envigo) (lines 102 and 103 in the revised version). We also have added a table with the composition of this diet as Supplementary Material: Table S1 (lines 109-110 in the revised version).

Comment: Fructose-Manufacture

Answer: Done. Fructose was from Sigma-Aldrich, (line 105 in the revised version).

Comment: Fat – what source of fat? Lard? Must be specified. I recommend that authors add a table of the diets used as a supplement material

Answer: Thank you for the recommendation. High-fat diet (Teklad Diet TD.06414, Envigo) is rich in lard (line 104 in the revised version).  We also have added a table with the composition of this diet as Supplementary Material: Table S2 (lines 109-110 in the revised version).   

Comment: How were the probiotics administered during the experimental period? Must be specified!

Answer: As suggested by the reviewer, this part has now been clarified in the revised manuscript as follows; “the probiotic was administrated at a daily dose of 1x109 CFU per animal mixed in the experimental diet” (lines 110 and 111 in the revised version).

Comment: Experimental period was 10 weeks + 2 weeks acclimatization. Please, correct line 108 to 10 weeks.

Answer: Done. (Line 112, in the revised version).

Comment: Line 112, centrifugation, specify rpm, give name and manufacture of the centrifuge. What temp?

Answer: As suggested by the reviewer, this part has now been clarified in the revised manuscript as follows; “Serum was obtained from blood samples after centrifugation at a maximum speed of 3.000 rpm for 15 min at 4ºC in a Beckman coulter centrifuge” (lines 115-117, in the revised version).

Comment: What is physiological serum (line 113)? Do authors mean saline or PBS? Must be clarified.

Answer: The reviewer is correct; we have changed the word serum to saline (line 118 in the revised version).

Comment: Line 135. Tissues were dehydrated. Which solution you used for dehydration? Was is xylen or methylbensoat?

Answer: Many thanks to reviewer for his observation.  Tissues were dehydrated with xylene (line 140 in the revised version).

Comment: Line 137, what is WAT? Please, give the full name before abbreviation.

Answer: As suggested by the reviewer, we have included the expanded full name of white adipose tissue (WAT) (line 142 in the revised version).

Comments (B): How did you collect and kept the stool samples? Were fecal samples collected in aseptic condition? During dissection? Before the dissection etc…

Answer: Thanks again for this additional request. Fecal samples were collected at the end of the study before the dissection of the mice, using an aseptic technique and immediately frozen at -80ºC for metagenomic analyses. This part has now been clarified in the revised manuscript (lines 159-160 in the revised version).

Comment: Line 149. What the housekeeping gene is encoding?

Answer: Housekeeping gene RPLPO encodes a ribosomal protein lateral stalk subunit PO. This part has now been clarified in the revised manuscript (lines 154-155 in the revised version).

Comment: Please, be consistent throughout the text body: If it CD68, CD44, TLR4, IL5 etc, use capital letters since it is abbreviations! Please, correct the same in table 1.

Answer: We agree with the reviewer´s assessment. Accordingly, throughout the revised manuscript and Table 1 of the Material and Methods section, we have corrected them using capital letters.

Comment: Line 155. (n=6). Is this number per group? Please, specify.

Answer: Following the reviewer´s suggestion, (n=6) has now been replaced by (n=6 mice per group) (line 162 in the revised version).

Comment: Line 169. Glucose evolution… Please, rephrase.

Answer: Thank you for this suggestion. “Glucose evolution” has been rephrased by “serum glucose levels over time” (line 177 in the revised version).

Comment: Line 177. PCoa – must be PCoA.

Answer: Done. “PCoa” has been amended to “PCoA” (line 185 in the revised version).

RESULTS

Comment: Headings of result section sound like discussion/conclusion. I recommend to change to: Effects on body weight gain and serum lipids etc! Please change the headings throughout result section, e.g. 3.2. use ‘Effects on glucose metabolism’ etc.

Answer: We followed the reviewer´s recommendations, and we are grateful for these suggestions. Headings have now been modified in the result section accordingly (lines 194, 211, 230, 273, 308, 331, and 354, respectively, in the revised version).

Comment: Line 227, 229. Are you present you result in thousand? May be punctuation is not necessarily… 4.529 = 4 529? Please check.

Answer: Many thanks to the reviewer for this observation. Now, 4.529, etc., have been replaced in the revised manuscript by 4,529 (lines 234-236 in the revised version).

Comment: Line 247, different text size of text for ‘Chi313 and Retnla’?

Answer: We are sorry about the different text sizes for CHI313 and RETNLA. It has been amended (line 255 in the revised version).

Comment: Figure 5 legend: Add numbers of animals per each group.

Answer: Done. The number of animals per each group (n=6 mice per each group) has now been added to the Figure 5 legend (line 330 in the revised version).

Comment: Heading 3.6 change to: Effect on gut microflora.

Answer: Done. “Lactiplantibacillus plantarum DSM 20174 modifies bacterial composition in the gut microbiota of HFHF fed mice” has now been substituted by “Effect of Lactiplantibacillus plantarum DSM 20174 on gut microflora” (line 331 in the revised manuscript).

Comment: Please, describe how many OTU has been observed assigned to different taxonomic levels.

Answer: Of a total of 1,644 sequenced OTUs, 59 were assigned to the species level, 438 to the genus level, 1,203 to the family level, 1,593 to the order level, 1,637 to the class level, and 1,643 to the phylum level.

Comment: I would like to see the OTU table as a supplement material (L2, L6). Will the sequencing data be available at pubmed database?

Answer: We have included the OTU table as Supplementary Material Table S3 (line 171 in the revised manuscript). For the moment, the sequencing data have not yet been deposited in a database due to confidential reasons. However, these data can be available from the authors upon reasonable request.

Comment: Since study has been focused on induced obesity, the ratio of Firmicutes to Bacteroidetes (F/B) would be important to show.

Answer: We have calculated the Firmicutes / Bacteroidetes ratio (F/B) (Bacillota/Bacteroidota) in the three groups studied (ND = 0.47±0.24; HFHF = 1.52±1.06 and HFHF+L.p = 1.44±1.09). The results are shown in the revised manuscript, lines 315-316.

Comment: Heading 3.7. please reconsider name.

Answer: Done. “Potential relations between underlying factors contributing to NAFLD and gut microbiota changes in HFHF fed mice” has now been substituted by “Association of some microbial taxa with underlined factors contributing to NAFLD” (line 354 in the revised manuscript).

DISCUSSION

Comment: You can add some heading directing readers to follow your discussion.

Answer: Thank you for this suggestion. We have added headings to the discussion section on lines 398 and 436 in the revised manuscript.

Comment: Line 384. Instead of ameliorate glucose and lipid homeostasis. I would recommend to use … glucose and lipid metabolic disturbances…   

Answer: Thank you for this suggestion. In the revised manuscript, we have changed “ameliorate glucose and lipid homeostasis to ”ameliorate glucose and lipid metabolic disturbances” (line 392).
